# A Method Based on Timing Weight Priority and Distance Optimization for Quantum Circuit Transformation

**DOI:** 10.3390/e25030465

**Published:** 2023-03-07

**Authors:** Yang Qian, Zhijin Guan, Shenggen Zheng, Shiguang Feng

**Affiliations:** 1School of Information Science and Technology, Nantong University, Nantong 226000, China; 2Peng Cheng Laboratory, Shenzhen 518055, China; 3School of Computer Science and Engineering, Sun Yat-sen University, Guangzhou 510275, China

**Keywords:** quantum circuit transformation, qubit mapping, subgraph isomorphism, heuristic optimization

## Abstract

In order to implement a quantum circuit on an NISQ device, it must be transformed into a functionally equivalent circuit that satisfies the device’s connectivity constraints. However, NISQ devices are inherently noisy, and minimizing the number of SWAP gates added to the circuit is crucial for reducing computation errors. To achieve this, we propose a subgraph isomorphism algorithm based on the timing weight priority of quantum gates, which provides a better initial mapping for a specific two-dimensional quantum architecture. Additionally, we introduce a heuristic swap sequence selection optimization algorithm that uses a distance optimization measurement function to select the ideal sequence and reduce the number of SWAP gates, thereby optimizing the circuit transformation. Our experiments demonstrate that our proposed algorithm is effective for most benchmark quantum circuits, with a maximum optimization rate of up to 43.51% and an average optimization rate of 13.51%, outperforming existing related methods.

## 1. Introduction

Quantum computing, a new computing paradigm that leverages the superposition and entanglement features of quantum mechanics, has the potential to solve many problems faster than classical computers, such as integer factorization [1] and linear equations [2]. Currently, quantum computing has entered the era of noisy intermediate-scale quantum (NISQ) devices [3]. In contrast to the idealized quantum circuit model, physical quantum architectures have a connectivity constraint, limiting the set of allowable two-qubit gates between specific pairs of qubits. To ensure that a quantum circuit is functionally equivalent to the desired computation and satisfies the connectivity constraint, quantum circuit transformations must be applied. However, NISQ devices suffer from crosstalk noise, which results from unexpected interactions or uncorrected control of qubits. This noise compromises the fidelity of the final circuit execution results [4]. Therefore, minimizing the number of SWAP gates added to the circuit during the transformation process is a critical task for circuit optimization.

The process of transforming a quantum circuit can be divided into two sub-procedures. The first involves finding an initial mapping, which maps logical qubits to physical qubits. The second sub-procedure involves handling quantum gates that violate the interaction constraint in the current mapping by inserting SWAP gates to lead to neighboring qubits. However, determining the minimal number of SWAP gates required for the quantum circuit transformation is an NP complete problem [5]. To solve this problem, various methods have been proposed [6,7,8,9,10,11,12,13,14,15]. One type of method formulates the problem mathematically and uses solvers such as integer linear programming [16], satisfiability module theory [4], constraint planning [17], or Boolean solvers [18] to find solutions. These methods are effective for small quantum circuits. The second type of method uses heuristic algorithms to solve the problem. For example, the backward traversal-based mapping method SABRE [6], Monte Carlo tree search [8], A* search algorithm [7,19], good initial mapping generation [9], Bridge gate insertion [10], reversibility-based comparison of forward and reverse circuit transformation processes [11], greedy algorithm [12], simulated annealing [13], subgraph isomorphism-based mapping [14], and an algorithm based on dynamic look-ahead heuristic cost functions [15] have been proposed. These heuristic algorithms use different evaluation functions to determine the best mapping strategy, based on various factors such as the number of two-qubit quantum gates, the depth of the circuit, the distance between the control qubit and the target qubit, or the topology of the quantum circuit. Some methods also consider the dependency of quantum gates in the circuit and prioritize the execution of preceding gates.

In this paper, we follow a similar approach to the literature [14] by using the number of additional CNOT gates as a metric for measurement, and we utilize subgraph isomorphism as the basic algorithm in the initial mapping. However, in addition to this, we consider the execution order of the quantum gates in the logical quantum circuit during the initial mapping process. Furthermore, we propose a forward-looking heuristic algorithm during the routing process to compare the quantum gate interaction distances, particularly in cases where the interactability rates are equal, to find a better routing path. The proposed method achieves the final transformation of logical quantum circuits to executable physical quantum circuits.

The process of transforming a quantum circuit can be divided into two main steps: (i) initial qubit mapping and (ii) insertion of SWAP gates. In the initial qubit mapping step, we first assign a timing weight to each edge in the circuit’s interaction graph and then use a subgraph isomorphism algorithm based on these timing weights to find an initial mapping between the logical qubits and physical qubits. In the SWAP gates insertion step, there may be several possible sequences of SWAP gates that can be applied. To determine the best sequence, we define an interactivity value and a distance optimization measure for each sequence. We propose a sequence selection optimization algorithm based on maximizing the distance measure optimization function, which selects the SWAP gates sequence with the highest value. The experimental results demonstrate the effectiveness of our algorithm.

The paper is structured as follows. In Section 2, we provide definitions and notations related to quantum gates and circuits. Section 3 presents our initial mapping algorithm based on timing weight subgraph isomorphism. In Section 4, we propose a SWAP gate sequence selection optimization algorithm for quantum circuit transformation. In Section 5, we evaluate our approach on benchmark quantum circuits and compare it with a state-of-the-art method. We conclude the paper in Section 6.

## 2. Preliminaries

In this section, we provide the fundamental definitions and notations related to quantum circuits.

### 2.1. Quantum Gate and Quantum Circuit

In quantum computing, quantum bits (qubits) are the fundamental unit of quantum information [20]. In contrast to classical bits, which only have two states (0 and 1), the state |ϕ〉 of a qubit is a superposition of the two states |ϕ〉=α|0〉+β|1〉, where ${\left|\alpha \right|}^{2}+{\left|\beta \right|}^{2}=1$. Quantum gates are used to operate on qubits in quantum computers. Single-qubit gates operate on a single qubit, while two-qubit gates operate on two qubits. A two-qubit gate *g* is denoted as g=〈p,q〉 and indicates that *g* operates on qubits *p* and *q*. The CNOT gate (see Figure 1a) is an example of a two-qubit gate, where *p* is the control qubit and *q* is the target qubit. If *p* is in state 1, the CNOT gate flips the state of *q*, and if *p* is in state 0, *q* remains unchanged. A SWAP gate 〈p,q〉 exchanges the states of *p* and *q*, and can be realized by cascading three CNOT gates (see Figure 1b).

Quantum circuits are used to describe quantum algorithms and are composed of qubits, quantum gates, measurement gates, classical registers, and so on. We use LC=(Q,C) to denote a quantum circuit, where *Q* and *C* represent the set of qubits and the set of quantum gates in the circuit, respectively. Since the CNOT gate and all single-qubit gates are widely used as universal quantum gates set, and the single-qubit gates are compliant with the connectivity constraint, we only consider quantum circuits composed of two-qubit gates in this paper.

**Example 1.** 
*Figure 2 depicts a quantum circuit LC=(Q,C), where $Q=\{q_{0},q_{1},q_{2},q_{3}\}$ and $C=\{g_{0}=\langle q_{2},q_{0}\rangle ,g_{1}=\langle q_{3},q_{2}\rangle ,g_{2}=\langle q_{0},q_{3}\rangle ,g_{3}=\langle q_{0},q_{2}\rangle ,g_{4}=\langle q_{3},q_{2}\rangle ,g_{5}=\langle q_{0},q_{3}\rangle ,g_{6}=\langle q_{3},q_{1}\rangle ,g_{7}=\langle q_{0},q_{1}\rangle \}$, consisting of four qubits and eight CNOT gates.*


### 2.2. Dependency Graph and Interaction Graph

Given a quantum circuit LC=(Q,C), we define two graphs: the dependency graph, DG, and the interaction graph, IG. DG is a directed acyclic graph whose nodes are the gates in *C*. There is a directed edge from gate $g_{i}$ to gate $g_{j}$ in DG if $g_{i}$ operates on a qubit *q* and $g_{j}$ is the next gate that operates on *q* after $g_{i}$. IG is an undirected graph whose nodes are the qubits in *Q*. There is an edge between two qubits $q_{i}$ and $q_{j}$ in IG if they are operated by some gate. Figure 3 shows the dependency graph and interaction graph of the quantum circuit in Figure 2. The dependency graph captures the dependencies between gates in the circuit, while the interaction graph shows the connectivity of qubits in the circuit.

The dependency graph is a useful tool to represent the execution order of quantum gates in a quantum circuit, while the interaction graph is commonly used to represent the interaction relationships between qubits in a quantum circuit [14].

## 3. Initial Qubit Mapping

A physical quantum device can be represented by its coupling graph CG, which is an undirected graph (V,E) where each qubit in the device is a node in *V*, and there is an edge $(q_{i},q_{j})\in E$ between two nodes $q_{i}$ and $q_{j}$ if they can be operated by a two-qubit gate in the device. For instance, Figure 4 shows the coupling graph of IBM QX20 Tokyo.

Given a quantum circuit LC=(Q,C) and a coupling graph CG=(V,E), an initial mapping π is an injection from *Q* to *V*. To find an effective initial mapping, it is important to consider the following two conditions when designing the algorithm:Logical qubits that interact frequently should be mapped to physically adjacent qubits.Qubits with high execution priority, based on the order of quantum gates in the circuit, should be mapped to adjacent physical qubits.

By taking into account these conditions, we can design an initial mapping algorithm that can improve the performance and efficiency of quantum circuit execution on physical devices.

### 3.1. Timing Weight

To further improve the effectiveness of the initial mapping algorithm, we can introduce another metric called timing weight. The timing weight takes into account the execution order of quantum gates and the interaction relationships between qubits in the quantum circuit. In this subsection, we define the timing weight for quantum gates and edges in the interaction graph and explain how it can be used to design an improved initial mapping algorithm.

**Definition 1.** 
*Given a quantum circuit LC=(Q,C), where $Q=\{q_{0},q_{1},\cdots ,q_{m}\}$ and $C=\{g_{0},\cdots ,g_{n-1}\}$, the timing weight, $k_{i}$, for each quantum gate $g_{i}$ is*

(1)
$$k_{i}=n-i,$$

*where 0≤i≤n−1 is the execution order of $g_{i}$.*


**Definition 2.** 
*Let IG be the interaction graph of LC. For each edge (p,q) in IG, where p,q are qubits in LC, the timing weight, $\omega_{\left(p,q\right)}$, of (p,q) is*

(2)
$$\omega_{\left(p,q\right)}=\sum_{g_{i}=\langle p,q\rangle }k_{i}+\sum_{g_{j}=\langle q,p\rangle }k_{j},$$

*where $k_{i}\left(k_{j}\right)$ is the timing weight of $g_{i}\left(g_{j}\right)$.*


The timing weight measures the importance of a quantum gate in the execution order of a quantum circuit, with lower weights assigned to gates executed later. The timing weight of an edge in the interaction graph represents the total timing weight of the gates that interact through that edge.

**Example 2.** 
*Figure 5 displays the timing weights of the gates in the circuit of Figure 2 and the timing weights of the edges in its interaction graph. The edge $\left(q_{0},q_{2}\right)$ has the largest weight, indicating that the gates associated with these qubits are executed first in the circuit. As $q_{0}$ and $q_{2}$ contain the most CNOT gates in the circuit, they should be mapped as nearest neighbors.*


### 3.2. Initial Mapping Based on Timing Weight

To construct an initial mapping for a quantum circuit LC and a coupling graph CG, we first compute the interaction graph IG of LC with timing weights assigned to each edge. If IG is isomorphic to a subgraph of CG, we can construct an initial mapping from the isomorphism without adding any SWAP gates. However, when IG cannot be directly mapped to CG, we use a partial subgraph isomorphism based on timing weight. This means that edges with larger timing weights are given higher priority in the mapping process.

The timing weight of each edge in IG not only indicates the qubits with more CNOT executions but also the execution order of the CNOT gates in the circuit. Thus, it is an important factor in finding an optimal initial mapping that provides a better precondition for subsequent quantum circuit transformations. We propose an initial mapping algorithm based on timing weight subgraph isomorphism, as outlined in Algorithm 1. The algorithm takes LC and CG as inputs and constructs an initial mapping by searching every edge from the one with the highest timing weight.

Algorithm 1 outlines the steps for constructing an initial mapping based on timing weight subgraph isomorphism. To begin, we initialize the set of edges $Edge_{s}et$ of IG. Lines 2–8 compute the interaction graph, IG, of LC with timing weights for all edges. Line 9 sorts the edges from the largest timing weight to the smallest timing weight. Lines 10–16 construct a graph ig starting from the edge with the largest timing weight and add the edge to ig if there is a subgraph isomorphism from ig to CG. If there is no subgraph isomorphism, the edge is skipped. This process is repeated until all the edges in $Edge_{s}et$ are traversed. Finally, an initial mapping is constructed from ig and saved in results. It is important to note that the initial mapping obtained may not be unique, as it depends on the subgraph isomorphism found from ig to CG in the algorithm.
**Algorithm 1** Initial mapping based on timing weight subgraph isomorphism**Input:** A quantum circuit LC=(Q,C) and a coupling graph CG=(V,E).**Output:** An initial mapping from *Q* to *V*.1:**Initialize:** results←∅; Edge_set←∅2:**for** each gate $g_{i}\in LC$ **do**3:    $k_{i}\leftarrow \left|C\right|-i$4:    **if** $g_{i}=\langle p,q\rangle$ or $g_{i}=\langle p,q\rangle$ **then**5:          $\omega_{\left(p,q\right)}\leftarrow sum\left(k_{i}\right)$6:          $Edge\_set\leftarrow [\left(p,q\right),\omega_{\left(p,q\right)}]$7:    **end if**8:**end for**9:Edge_set.sort()10:**while** Edge_set≠∅**do**11:    **if** ig∪Edge_set.pop() is isomorphic to a subgraph of CG **then**12:          ig←ig∪Edge_set.pop()13:    **else**14:        skip(Edge_set.pop())15:    **end if**16:**end while**17:results←mapping(ig,CG)18:**return** results

**Example 3.** 
*By applying Algorithm 1 to the quantum circuit shown in Figure 2 and the coupling graph of IBM QX20 shown in Figure 4, we can obtain three initial mappings (colored in red) as shown in Figure 6. It is important to note that the initial mapping obtained may not be unique, as it depends on the subgraph isomorphism found from the algorithm.*


## 4. Quantum Circuit Transformation and Optimization

Given the quantum circuit LC, the coupling graph CG, and the mapping π from LC to CG, we need to ensure that every quantum gate in LC is mapped to adjacent qubits in CG. If a gate violates this interaction condition, we can add SWAP gates to the circuit to make the two qubits operated by the gate adjacent. This process can be repeated for every edge in CG. We use SWAP(e) to denote the application of a SWAP gate on the two ends of *e*. After performing a swap operation, a new mapping $\pi^{\prime }$ can be obtained by updating π. For instance, in Figure 6, the initial mapping $\pi_{1}$ is $\left\{q_{0}\to 1,q_{1}\to 0,q_{2}\to 10,q_{3}\to 6\right\}$. We can apply SWAP(5,6) and SWAP(1,6) to obtain the mapping $\pi_{2}=\left\{q_{0}\to 6,q_{1}\to 0,q_{2}\to 10,q_{3}\to 5\right\}$.

### 4.1. Swap Sequence Selection

During the quantum circuit transformation process, a sequence of SWAP gates is added to ensure that every gate in the circuit satisfies the interaction condition with respect to the coupling graph. However, this sequence is not unique, and we aim to find the shortest sequence that minimizes the number of gates in the resulting circuit.

Let π be a mapping from the quantum circuit LC to the coupling graph CG, and let
$$a=(SWAP\left(e_{1}\right),SWAP\left(e_{2}\right),\cdots ,SWAP\left(e_{n}\right))$$
be a sequence of SWAP gates, where each $e_{i}\left(1\leq i\leq n\right)$ is an edge of CG. We use $g_{ex}$ to denote the number of gates in LC that are executable after applying the sequence *a* from mapping π. To measure the effectiveness of the transformation, we define the interactivity value:(3)$$G_{val}\left(\pi ,a\right)=\frac{g_{ex}}{len(a)},$$
where len(a)=n is the number of SWAP gates in *a*. A larger value of $G_{val}\left(\pi ,a\right)$ indicates that the sequence *a* is more effective in transforming the circuit. Therefore, we aim to find the sequence *b* that maximizes $G_{val}\left(\pi ,b\right)$ for a given mapping π.

Now let us consider an example to see how the swap sequence selection algorithm works in practice.

**Example 4.** 
*Consider the quantum circuit shown in Figure 2 and the initial mapping $\pi_{1}=q_{0}\to 1,q_{1}\to 0,q_{2}\to 10,q_{3}\to 6$ in Figure 6. The first quantum gate that violates the interactive constraint is $g_{0}=\langle q_{2},q_{0}\rangle$. To address this issue, we explore four different swap ways and the number of gates that can be executed for each one, as follows:*
*1.* 
*SWAP(5,6),SWAP(1,6): seven gates can be executed;*
*2.* 
*SWAP(2,6),SWAP(10,6): seven gates can be executed;*
*3.* 
*SWAP(1,7),SWAP(10,6): four gates can be executed;*
*4.* 
*SWAP(0,1),SWAP(5,0): six gates can be executed.*

*We can then evaluate the validity of each sequence using the metric $G_{val}$, and obtain $G_{val}\left(\pi_{1},1\right)=7/2$, $G_{val}\left(\pi_{1},2\right)=7/2$, $G_{val}\left(\pi_{1},3\right)=2$, and $G_{val}\left(\pi_{1},4\right)=3$. Based on these values, we conclude that the candidate sequences are (i) and (ii).*


### 4.2. Sequence Selection Optimization

For large-scale quantum circuits, there are often multiple swap sequences that satisfy the interactivity constraint, as shown in Example 4. In this subsection, we propose a sequence selection optimization algorithm to find a better swap sequence from the candidate sequences.

Given the quantum circuit LC, the coupling graph CG, and the mapping π from LC to CG, we use $dist_{ph}\left(g,\pi \right)$ to denote the length of the shortest path between π(p) and π(q) in CG for every gate g=〈p,q〉 in LC. Let *a* be a sequence of SWAP gates, and let π[a] denote the new mapping obtained by applying *a* to π. We define the set $S=\{g|dist_{ph}\left(g,\pi \left[a\right]\right)<dist_{ph}\left(g,\pi \right)\}$, which contains the gates whose execution order is affected by the sequence *a*. We then define the distance optimization measure as follows:(4)$$\Omega_{g}\left(\pi ,a\right)=\sum_{g_{i}\in S}k_{i},$$
where $k_{i}$ is the timing weight of $g_{i}$.

The value of $\Omega_{g}\left(\pi ,a\right)$ depends on the timing weight of every gate in *S*. A gate $g_{i}$ with a large value of $k_{i}$ means that $g_{i}$ has a high execution order. Therefore, $\Omega_{g}\left(\pi ,a\right)$ indicates the priority of the sequence *a*; a sequence with a larger $\Omega_{g}\left(\pi ,a\right)$ should be selected first. To select the best swap sequence, we compute the $\Omega_{g}\left(\pi ,a\right)$ values for each candidate sequence using Equation (Equation 4), and choose the sequence with the largest value. The swap sequence selection optimization algorithm is presented in Algorithm 2. The inputs of the algorithm are a quantum circuit, a coupling graph, and an initial mapping. The output is a transformed quantum circuit that is compliant with the connectivity constraint.
**Algorithm 2** Sequence selection optimization algorithm**Input:** A quantum circuit LC, a coupling graph CG, and an initial mapping $\pi_{0}$.**Output:** A quantum circuit PC that is compliant with connectivity constraint of CG.1:**Initialize:** swap←∅; a←∅; $\pi \leftarrow \pi_{0}$; PC←∅2:**for** each gate g=〈p,q〉∈LC **do**3:    **if** $dist_{ph}\left(g,\pi \right)=1$ **then**4:        LC←LC.remove(g)5:        PC←PC.add(g)6:    **else**7:      $\max G_{val}\leftarrow G_{val}\left(\pi ,\left(a,SWAP\right)\right).sort\left(\right)$8:        (compute the $G_{val}$ for all possible SWAP gates and save the largest ones)9:        **for** each $SWAP\in \max G_{val}$ **do**10:           $swap\leftarrow \max\Omega_{g}\left(\pi ,\left(a,SWAP\right)\right)$11:           (compute the $\Omega_{g}$ for all sequences in $\max G_{val}$ and take one of the largest)12:        **end for**13:        a←(a,swap)14:        π←π[swap]15:        PC←PC.add(sawp)16:        PC←PC.add(g)17:        LC←LC.remove(g)18:    **end if**19:**end for**20:**return** PC

The algorithm starts by initializing the necessary variables. It adds all quantum gates that meet the connectivity constraint in PC and removes them from LC according to their execution order in the current mapping, as described in Lines 3–5. If no gate can be added to PC, indicating that SWAP gates must be inserted, the algorithm computes the $G_{val}$ for all possible SWAP gate sequences and saves those with the largest value in $\max G_{val}$, as shown in Line 7. Next, Lines 9–12 compute the $\Omega_{g}$ for all sequences in $\max G_{val}$ and selects the one that has the largest value. Finally, Lines 13–17 update LC, PC, π, and *a* based on the SWAP gates added.

To illustrate how the sequence selection optimization algorithm works, let us consider the following example.

**Example 5.** 
*In Example 4, we compare the two SWAP gate sequences (i) and (ii), which have the same $G_{val}$ value. Using Equation (Equation 4), we obtain the following distances and cost values:*

$$\begin{array}{cc} \pi_{1}: & dist_{ph}\left(g_{0},\pi_{1}\right)=dist_{ph}\left(g_{3},\pi_{1}\right)=dist_{ph}\left(g_{6},\pi_{1}\right)=2\\ & dist_{ph}\left(g_{1},\pi_{1}\right)=dist_{ph}\left(g_{2},\pi_{1}\right)=dist_{ph}\left(g_{4},\pi_{1}\right)=1\\ & dist_{ph}\left(g_{5},\pi_{1}\right)=dist_{ph}\left(g_{7},\pi_{1}\right)=1\;\\ \pi_{1}\left[\left(i\right)\right]: & dist_{ph}\left(g_{i},\pi_{1}\left[\left(i\right)\right]\right)=1\;\left(0\leq i\leq 6\right),dist_{ph}\left(g_{7},\pi_{1}\left[\left(i\right)\right]\right)=2\\ & \Omega_{g}\left(\pi_{1},\left(i\right)\right)=8+5+2=15\;\\ \pi_{1}\left[\left(ii\right)\right]: & dist_{ph}\left(g_{i},\pi_{1}\left[\left(ii\right)\right]\right)=1\;\left(0\leq i\leq 5\right),dist_{ph}\left(g_{6},\pi_{1}\left[\left(ii\right)\right]\right)=2,\;\;\;\;\;\;\;\;\;\;\;\;\;\;\;\;\;\;\;\;\;\;\;\;\;\;\;\;\;\;\;\;\;\;\;\;\;\;\;\;\;\;\;\;\;\;\;\;\;\;\;\;\;\;\;\;\;\;\;\;\;\;\;\;\;\;\;\;\;\;\;\;\;\;\;\;\;\;\;\;\;\;\;\;\;\;\;\;\;\;\;\;\;\;\;\;\;\;\\ & dist_{ph}\left(g_{7},\pi_{1}\left[\left(ii\right)\right]\right)=1\\ & \Omega_{g}\left(\pi_{1},\left(ii\right)\right)=8+5=13 \end{array}$$

*Since both sequences have the same $G_{val}$ value, we choose the sequence with the higher values according Algorithm 2. Therefore, we choose sequence (i) as the optimal swap sequence.*


## 5. Evaluation

To assess the effectiveness of the proposed methods, we now turn to evaluate their performance on a variety of quantum circuits. We provide experimental results and comparisons with existing approaches in this section.

The proposed methods were implemented in Python, and the experiments were conducted on a Windows 10 machine with an Intel Core i7 processor and 16 GB of RAM. The quantum circuits used in the experiments were publicly available benchmarks evaluated in [14]. The experimental comparison baseline is the number of CNOT gates inserted by the quantum circuit transformation. In this paper, we classify quantum circuits with less than 1000 gates as small scale, those with greater than 1000 and less than 10,000 gates as medium scale, and those with more than 10,000 gates as large scale.

Table 1, Table 2 and Table 3 show the results of the experiments for small scale, medium scale, and large scale circuits, respectively. The first column in each table lists the benchmark, the second column “QubitNu” shows the number of qubits in the circuit, and the third column “GateNu” shows the number of CNOT gates in the circuit. The fourth and fifth columns, labeled “AuxGt”, show the number of auxiliary CNOT gates added by the transformation algorithm in [6,14], respectively, and serve as the main comparative baseline for the experiments. The sixth column shows the number of auxiliary CNOT gates added by the algorithm proposed in this paper. The seventh and eighth columns, labeled “Comp.”, show the optimization rate compared with the literature [6,14]. The last row of each table shows the average optimization rate.

In order to evaluate the effectiveness of the proposed quantum circuit transformation optimization algorithm, we compare the experimental results with two previous algorithms, Sabre [6] and Topgraph [14]. Sabre is integrated in the IBMQ quantum cloud platform, which represents the current optimal level of IBMQ, and Topgraph is the basis of algorithm improvement in this paper. Therefore, they are selected as the baselines for the experiments in this paper. From the experimental results in the three tables, it is evident that the quantum circuit transformation optimization algorithm in this paper outperforms Sabre, with the highest average optimization rate reaching 70.15%. In comparison with Topgraph, the algorithm proposed in this paper achieves a minimum average optimization rate of 4.30% on small-scale circuits and 21.79% on large-scale circuits. Although two quantum circuits in the medium-scale circuit are not optimized, the average optimization rate still reaches 22.04%. Overall, the proposed algorithm has a significant optimization effect on circuits of any scale, with the larger circuits showing more obvious optimization.

## 6. Conclusions

In this paper, we proposed a timing-based optimization algorithm for quantum circuit transformation that defines a timing weight for every edge in the interaction graph of the circuit. By using a subgraph isomorphism algorithm based on timing weights, we can obtain a high-quality initial mapping. We also proposed a sequence selection optimization algorithm based on the distance measure optimization function for selecting the best SWAP gate sequence. The experimental results show that our algorithm outperforms the existing benchmarks for arbitrary scale quantum circuits, demonstrating its effectiveness. However, there are other factors that can affect the execution of quantum computation, such as quantum gate execution errors and qubit coherence time, which could be incorporated into a study for comprehensive consideration. Our future research direction will be to adapt the algorithm in this paper to account for these factors.

## Figures and Tables

**Figure 1 entropy-25-00465-f001:**
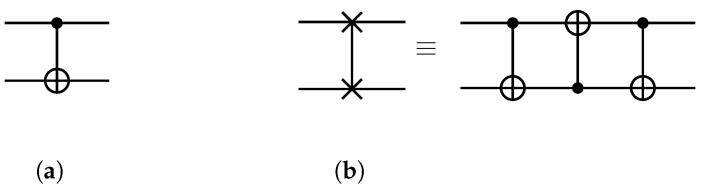
The CNOT gate and the SWAP gate. (**a**) A CNOT gate. (**b**) A SWAP gate and its realization by CNOT gates.

**Figure 2 entropy-25-00465-f002:**
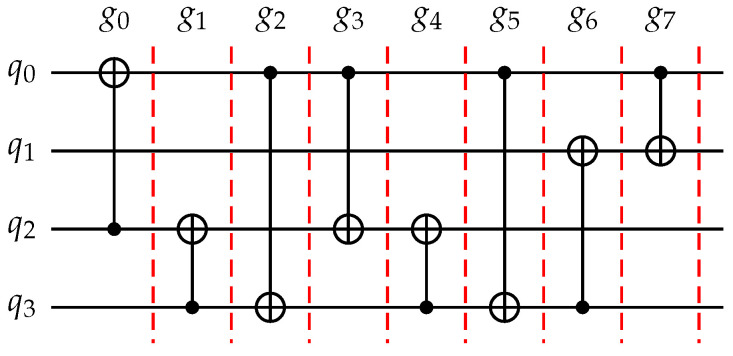
A quantum circuit with four qubits and eight CNOT gates.

**Figure 3 entropy-25-00465-f003:**
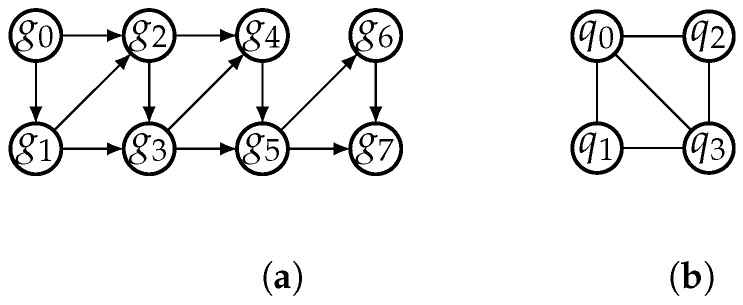
The dependency graph and interaction graph of the quantum circuit in Figure 2. The dependency graph, shown in (**a**), is a directed acyclic graph with nodes representing gates in the circuit and directed edges representing dependencies between gates. The interaction graph, shown in (**b**), is an undirected graph with nodes representing qubits in the circuit and edges representing interactions between qubits.

**Figure 4 entropy-25-00465-f004:**
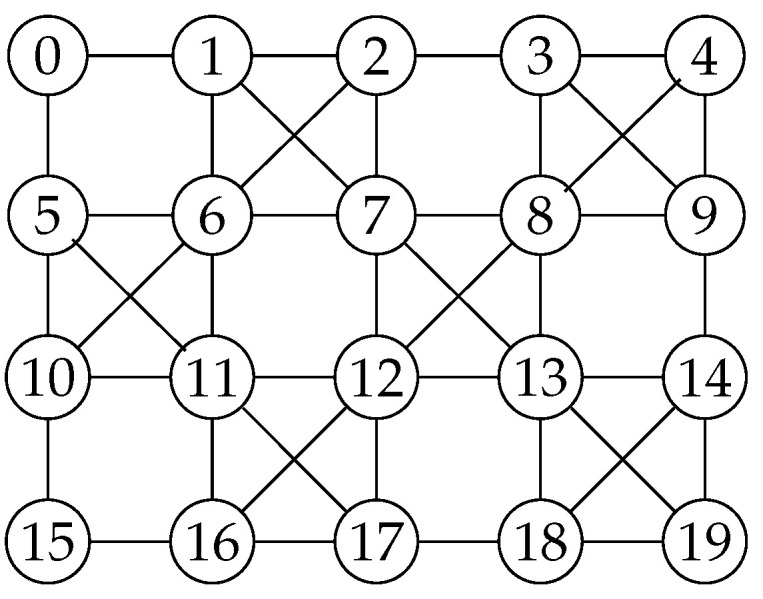
The coupling graph of IBM QX20 Tokyo.

**Figure 5 entropy-25-00465-f005:**
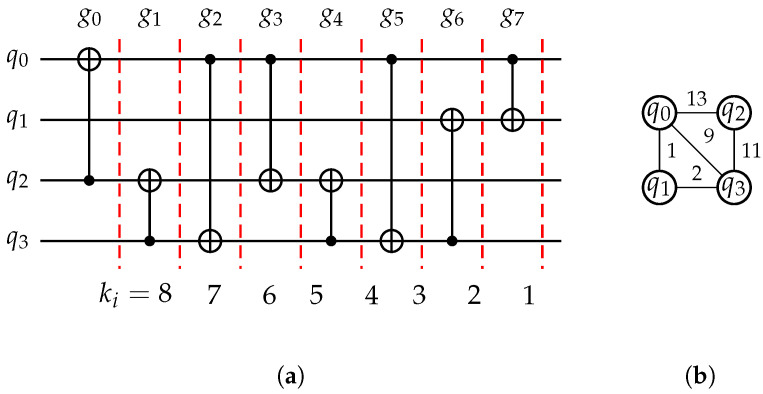
The timing weights of gates and edges. (**a**) The timing weight of each gate in the circuit of Figure 2. (**b**) The timing weight of each edge in the interaction graph of the circuit.

**Figure 6 entropy-25-00465-f006:**
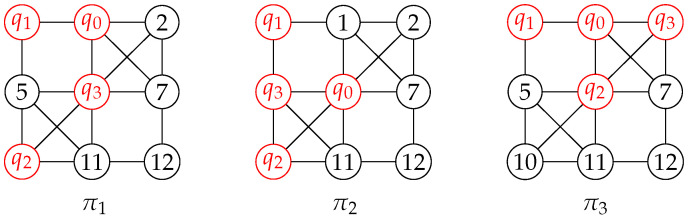
Three initial mappings from the circuit in Figure 2 to IBM QX20.

**Table 1 entropy-25-00465-t001:** The experimental results on small-scale circuits.

Benchmark			AuxGt in [6]	AuxGt in [14]	AuxGt	Comp.	
Circuit	QubitNu	GateNu				With [6]	With [14]
4mod5_v1_22	5	21	0	0	0	0.00%	0.00%
mod5mils_65	5	35	0	0	0	0.00%	0.00%
decod24v2_43	4	52	0	0	0	0.00%	0.00%
4gt13_92	5	66	0	0	0	0.00%	0.00%
qft_10	10	200	54	39	33	38.89%	15.38%
qft_16	16	512	186	153	117	37.10%	25.53%
ising_model_10	10	480	0	0	0	0.00%	0.00%
ising_model_13	13	633	0	0	0	0.00%	0.00%
ising_model_16	16	786	0	0	0	0.00%	0.00%
rd84_142	15	343	105	72	66	37.14%	8.33%
**Average**						11.31%	4.72%

**Table 2 entropy-25-00465-t002:** The experimental results on medium-scale circuits.

Benchmark			AuxGt in [6]	AuxGt in [14]	AuxGt	Comp.	
Circuit	QubitNu	GateNu				With [6]	With [14]
sym6_145	7	3888	1272	513	405	68.16%	21.05%
z4_268	11	3073	1365	630	363	73.41%	42.38%
radd_250	13	3213	1275	555	576	54.82%	−3.78%
cycle10_2_110	12	6050	2622	1194	969	63.04%	18.84%
adr4_197	13	3439	1614	630	648	59.85%	−2.86%
misex1_241	15	4813	1521	786	444	70.81%	43.51%
rd73_252	10	5321	2133	1095	732	65.68%	33.15%
square_root_7	15	7630	2598	1338	1017	60.85%	23.99%
**Average**						64.58%	22.04%

**Table 3 entropy-25-00465-t003:** The experimental results on large-scale circuits.

Benchmark			AuxGt in [6]	AuxGt in [14]	AuxGt	Comp.	
Circuit	QubitNu	GateNu				With [6]	With [14]
co14_215	15	17,936	8982	4257	2658	70.41%	37.56%
rd84_253	12	13,658	6147	2352	1827	70.28%	22.32%
sqn_258	10	10,223	4344	1578	1212	72.10%	23.19%
sym9_193	11	34,881	16,653	5589	5361	67.81%	4.08%
**Average**						70.15%	21.79%

## Data Availability

The data that support the findings of this study can be obtained from the authors upon reasonable request.

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
