# Peer review of "A Method Based on Timing Weight Priority and Distance Optimization for Quantum Circuit Transformation"

_entropy, 2023, doi:10.3390/e25030465_

Round 1
Reviewer 1 Report
Can be accepted
Author Response
We are grateful for your acceptance and endorsement of our manuscript. We sincerely apologize for any language issues that were present in our initial submission and acknowledge the need for improvement in our English writing skills. As such, we have thoroughly revised the entire paper and rewritten certain paragraphs to enhance the clarity of the manuscript.
Reviewer 2 Report
The manuscript describes an interesting quantum circuit transformation method and an interesting set of results.
However, the manuscript does not address a number of important aspects and has a few significant shortcomings.
The following issues need to be addressed:
[1] Literature review: on line 29 in introduction a block of 9 references is cited without any discussion of what these works are actually about. A short discussion on what is presented in these works is needed
[2] Literature review: references [14] to [19] do have a reasonable level of review and description - however it needs to be clearer how the present work relates to or is different from these works
[3] discussion of novelty - see point [2] - without detailing the differences and similarity of presented work with [14]-[19] (and also previous works [5]-[13]) it remains largely unclear what main contribution and novelty of proposed transformation method are
[4] Comparison against a single set of results without justifying why this is appropriate. All results are only compared a single previous work - i.e. Ref[18]. This assumes that Ref.[18] represents the absolute best benchmark solution set - this is not clear from the review and introduction. A comparison against a wider range of results is recommended.
Further, smaller comments:
[a] in the introduction (line 19) - first explain that in the quantum circuit model all qubits can access all other qubits, in contrast to real hardware. So, make it clearer why this type of transformation is needed for a wider audience
[b] In introduction, explain in more detail how noise in NISQ-era hardware and need for minimization of SWAPs are directly related - in its current form this is not clear to a wider audience with limited familiarity with Quantum Computing
Author Response
We would like to express our sincere gratitude for providing us with the opportunity to revise and improve our manuscript titled "A Method Based on Timing Weight Priority and Distance Optimization for Quantum Circuit Transformation." We greatly appreciate your insightful suggestions and comments, which have helped us significantly enhance the clarity and quality of our paper. We apologize for any language difficulties in our initial submission and acknowledge the need to improve our English writing skills. In order to meet the professional publication standards, we have carefully reviewed and revised the entire paper. We have also rewritten certain paragraphs to further improve the clarity and readability of the manuscript
We sincerely value the time and effort the reviewers have put into their work and hope that our revisions have addressed their concerns. A summary of the major corrections and responses to the your comments is provided below.
The manuscript describes an interesting quantum circuit transformation method and an interesting set of results. However, the manuscript does not address a number of important aspects and has a few significant shortcomings.
The following issues need to be addressed:
[1] Literature review: on line 29 in introduction a block of 9 references is cited without any discussion of what these works are actually about. A short discussion on what is presented in these works is needed.
Response:
We appreciate your suggestion to provide more details on the references cited in our introduction. We have carefully reviewed the literature and have made significant revisions to our manuscript accordingly. Specifically, in lines 33-36 of page 1, we now include a concise summary and discussion of the mathematical approach proposed in the cited works. In addition, in lines 37-48, we have provided a detailed description of the various heuristics employed in the literature and have summarized these methods to provide a clearer understanding of their contributions.
[2] Literature review: references [14] to [19] do have a reasonable level of review and description - however it needs to be clearer how the present work relates to or is different from these works.
[3] discussion of novelty - see point [2] - without detailing the differences and similarity of presented work with [14]-[19] (and also previous works [5]-[13]) it remains largely unclear what main contribution and novelty of proposed transformation method are.
Response:
We appreciate the reviewer's feedback on the clarity of how our work relates to previous research, and on the need to highlight the novelty of our proposed method. In response to these comments, we have made the following revisions:
P2, Line 37-48: We have provided a short summary of all the heuristic algorithms cited in the paper, clarifying the connections and differences between them.
P2, Line 49-57: We have not only re-summarized the literature we used in the paper, but also highlighted the correlations and differences between the methods used in this paper. Specifically, our method considers the execution order of the quantum gates of the logical quantum circuit during the initial mapping process and finds a better routing path using a forward-looking heuristic algorithm during the subsequent routing process. We compare the quantum gate interaction distances for the case of equal interactability rates, which are the main innovation points of this paper. We hope these revisions address the reviewer's concerns and improve the clarity of our manuscript.
[4] Comparison against a single set of results without justifying why this is appropriate. All results are only compared a single previous work - i.e. Ref. [18]. This assumes that Ref. [18] represents the absolute best benchmark solution set - this is not clear from the review and introduction. A comparison against a wider range of results is recommended.
Response:
Thank you for your valuable comments. We have taken into consideration your suggestion to include a wider range of benchmark results in our comparison. As such, we have added experimental results of the Sabre algorithm, which is recognized as a landmark quantum circuit mapping algorithm in the industry and is integrated into the IBM Q Quantum Cloud platform. Many of the papers cited in our manuscript also compare their results with Sabre, making it a suitable benchmark for our algorithm. We have updated the results comparison with Sabre in Section 5 and provided a detailed description of our experimental protocol and results in the conclusion. Our results demonstrate that the proposed algorithm outperforms the Sabre algorithm on various scales of quantum circuits, further validating the credibility of our approach. We have also updated the table of all experimental results to reflect this new comparison. Thank you for your insightful comments, which have helped us improve the quality of our manuscript.
Further, smaller comments:
[a] in the introduction (line 19) - first explain that in the quantum circuit model all qubits can access all other qubits, in contrast to real hardware. So, make it clearer why this type of transformation is needed for a wider audience
Response:
Thank you for the helpful feedback. We have included an explanation of the quantum circuit model in the introduction (P1, Line 19-23). Specifically, we now clarify that in the quantum circuit model, all qubits have the ability to access all other qubits, which is not the case in physical quantum hardware. We believe that this clarification will make it easier for a wider audience to understand the motivation behind the need for the proposed transformation method.
[b] In introduction, explain in more detail how noise in NISQ-era hardware and need for minimization of SWAPs are directly related - in its current form this is not clear to a wider audience with limited familiarity with Quantum Computing
Response:
This is an excellent suggestion. In order to provide a clearer understanding to a wider audience with limited familiarity with Quantum Computing, we have added a detailed explanation in P1, Line 23-27 about how noise affects the fidelity of quantum circuits, as well as relevant literature on noise studies. These changes will enhance the reader's understanding of the motivation behind our study on quantum circuit mapping.
In summary, we appreciate the helpful and insightful feedback provided by the reviewer. We have made extensive revisions to our manuscript to address the issues raised, including adding more details on the literature review, discussing the novelty of our work, providing a comparison against a wider range of results, and clarifying the relationship between noise in NISQ-era hardware and the minimization of SWAPs. We believe that these changes significantly improve the quality and clarity of our manuscript, and we thank the reviewer for their time and effort in providing such thorough and thoughtful feedback.

Reviewer 3 Report
Optimization of quantum circuits is very important for practical quantum computing. The authors propose and test optimization algorithm to effectively map a general quantum circuit to its practical implementation. Their algorithm solves the subgraph isomorphism problem based on timing weights of quantum gates. In the second part of the paper the algorithm for the optimization of the sequence of swap gates is also proposed. The description of the algorithms is quite clear and the presented results of benchmarking are rather convincing. I find the paper worth publishing in the present form after probably minor typo corrections
Author Response
Thank you for accepting and endorsing our manuscript. We appreciate your insightful comments and suggestions, which have been invaluable in helping us to revise and improve the paper. We apologize for any careless mistakes that may have caused confusion. We have carefully reviewed the entire manuscript and have made necessary corrections, including the spelling errors. Thank you for reminding us to pay attention to these details.
Round 2
Reviewer 2 Report
The revised manuscript represents a clear improvement over the original version and key issues were addressed.